# Amino Acid Transporters Are a Vital Focal Point in the Control of mTORC1 Signaling and Cancer

**DOI:** 10.3390/ijms22010023

**Published:** 2020-12-22

**Authors:** Yann Cormerais, Milica Vučetić, Scott K. Parks, Jacques Pouyssegur

**Affiliations:** 1Department of Molecular Metabolism, Harvard T.H. Chan School of Public Health, Boston, MA 02115, USA; 2Department of Medical Biology, Centre Scientifique de Monaco (CSM), 98000 Monaco, Monaco; milivuk85@gmail.com (M.V.); scott.parks@gmail.com (S.K.P.); 3CNRS, INSERM, Centre A. Lacassagne, Faculté de Médecine (IRCAN), Université Côte d’Azur, 06107 Nice, France

**Keywords:** amino acid transporters, LAT1, SNAT2, ASCT2, xCT, mTORC1, cancer, growth factors, nutrients

## Abstract

The mechanistic target of rapamycin complex 1 (mTORC1) integrates signals from growth factors and nutrients to control biosynthetic processes, including protein, lipid, and nucleic acid synthesis. Dysregulation in the mTORC1 network underlies a wide array of pathological states, including metabolic diseases, neurological disorders, and cancer. Tumor cells are characterized by uncontrolled growth and proliferation due to a reduced dependency on exogenous growth factors. The genetic events underlying this property, such as mutations in the PI3K-Akt and Ras-Erk signaling networks, lead to constitutive activation of mTORC1 in nearly all human cancer lineages. Aberrant activation of mTORC1 has been shown to play a key role for both anabolic tumor growth and resistance to targeted therapeutics. While displaying a growth factor-independent mTORC1 activity and proliferation, tumors cells remain dependent on exogenous nutrients such as amino acids (AAs). AAs are an essential class of nutrients that are obligatory for the survival of any cell. Known as the building blocks of proteins, AAs also act as essential metabolites for numerous biosynthetic processes such as fatty acids, membrane lipids and nucleotides synthesis, as well as for maintaining redox homeostasis. In most tumor types, mTORC1 activity is particularly sensitive to intracellular AA levels. This dependency, therefore, creates a targetable vulnerability point as cancer cells become dependent on AA transporters to sustain their homeostasis. The following review will discuss the role of AA transporters for mTORC1 signaling in cancer cells and their potential as therapeutic drug targets.

## 1. Introduction: mTORC1 and Nutrient Sensing

The mechanistic target of rapamycin complex 1 (mTORC1) is the master regulator protein kinase of anabolic cell growth and proliferation in eukaryotic cells. At the cellular level, mTORC1 senses pro-growth signals, in the form of exogenous growth factors, hormones and, importantly for the scope of this review, intracellular nutrients (e.g., amino acids, AAs) to promote a switch from catabolic to anabolic metabolism [1]. This master kinase promotes the synthesis of proteins, de novo nucleotides and lipids; the three major classes of biomolecules required for cell growth and proliferation [2]. In order to balance the growth reward with the associated risk of this metabolic transition, with respect to cellular energy, reducing equivalents, and nutrient availability, mTORC1 is tightly regulated by a sophisticated network of growth factors and nutrient sensing pathways. Nevertheless, this upstream regulatory network can malfunction and is indeed dysregulated in a diverse array of diseases including autoimmune, neurological and metabolic disorders, as well as cancer and neoplastic diseases [1].

Parallel regulatory axes exist upstream of mTORC1 that impinge on two systems of small G-proteins that reside at the surface of the lysosomes: (1) the small Ras-related G-protein, Rheb, that relays signals from growth factors/hormones/oncogenes and cellular stressors and (2) the Rag GTPases that sense intracellular AA availability (Figure 1). Importantly, signals from both AA and growth factors/hormones are required to promote a full activation of mTORC1.

Rheb, in its GTP bound form, is an essential activator of mTORC1 kinase activity (Figure 1). The tuberous sclerosis complex (TSC), protein complex comprised of the tumor suppressor proteins TSC1, TSC2, and TBC1D7 is the only known negative regulator of Rheb [3]. Within the complex, TSC2 possesses the GTPase activating protein (GAP) activity toward Rheb and integrates signals from diverse hormonal, environmental, and stress events from the PI3K/AKT, RAS/ERK, LKB1/AMPK and other signaling pathways [4]. The activity of TSC is controlled, at least in part, through multi-site phosphorylation on TSC2 that regulates its subcellular localization [5]. During cellular stress or withdrawal of growth factors, the TSC complex is recruited, through an unknown mechanism, to the lysosomal surface where it decreases Rheb-GTP levels and turns off mTORC1 signaling (Figure 1). Conversely, in the absence of stress, growth promoting signals lead to the Akt- and/or Erk-dependent phosphorylation of TSC2 resulting in the immediate release of the TSC complex from the lysosome allowing the accumulation of Rheb-GTP and therefore mTORC1 activation (Figure 1). Mutations in the TSC complex underlie neoplastic diseases such as tuberous sclerosis complex (TSC) and lymphangioleiomyomatosis (LAM), while such mutations are less frequent in sporadic cancers. However, numerous oncogenes (activated Ras, PI3K-CA) and tumor suppressors (PTEN, LKB1) are part of the upstream signaling pathways that regulate the TSC complex. As such, the TSC complex is abnormally inhibited in up to 80% of human cancers, resulting in a growth factor-independent activation of mTORC1 kinase in tumors [4].

The Rag GTPases act in parallel to Rheb and are responsible for controlling mTORC1 signaling in response to the intracellular presence of specific AAs (leucine and arginine, Figure 1) [6]. In mammalian cells, the Rag proteins function as heterodimers where RagA or RagB associate with RagC or RagD. The presence of AAs leads to the conversion of the RAG heterodimer to its active conformation in which RagA/B is loaded with GTP and RagC/D is loaded with GDP. Unlike Rheb, the Rag GTPases do not stimulate the mTORC1 kinase activity. Instead, when found in their active conformation, they bind to Raptor, an essential component of mTORC1, and promote its recruitment at the lysosomal surface where it can be activated by Rheb [6]. Since the Rag GTPases do not exhibit any lipid anchorage motif, their lysosomal localization is dependent on their interaction with another protein complex—the Ragulator (Figure 1). The Ragulator is a pentameric protein complex comprised of LAMTOR 1–5, where LAMTOR 1 carries the lipid modification that allows the Ragulator, the Rag GTPases and therefore mTORC1 to co-localize at the lysosomal surface. The nucleotide loaded status of the Rag GTPases are modulated by a complex upstream signaling network that includes leucine and arginine sensors, the V-ATPase, amino acid transporter(s) and factors with GAP or GTP exchange activity (GTP exchange factors, GEFs) towards the Rags (Reviewed in [7]). In addition to signals relaying the direct leucine and arginine availability, recent studies have demonstrated that the Rag GTPases also sense methionine and glutamine through their downstream metabolites S-adenosylmethionine and α-ketoglutarate respectively [8,9]. Importantly, two others small GTPases are involved in membrane trafficking as regulators of mTORC1 signaling in response to AAs. The first one, Rab1A, has been shown to recruit and activate mTORC1 on the Golgi in a Rag-independent fashion [10]. Meanwhile, the adenosine diphosphate ribosylation factor-1 GTPase (Arf1) was shown to promote the lysosomal translocation and activation of mTORC1 at the lysosomal surface, also independently of the Rag GTPases [11]. These two studies suggest a multi-hub model of mTORC1 regulation where different GTPases regulate distinctive subcellular activation of mTORC1 in response to different AAs.

Even though oncogenic-activating mutations can be found in the upstream pathways of the Rag GTPases [12,13,14,15], these events are relatively rare and most solid tumors remain sensitive to AAs withdrawal. It is possible that the difference in mutation frequency between the TSC–Rheb and the Ragulator–Rag GTPases axis is due to the fact that growing tumors experience phases where nutrients are scarce, due to an incomplete vasculature, and retention of the mTORC1 regulation by AAs allows metabolic adaptation to occur in cancer cells with respect to nutrient availability. This dependency of mTORC1 on AAs therefore creates a targetable vulnerability point as cancer cells become dependent on AA transporters to sustain their homeostasis. In the following sections of the review we discuss the role of AA transporters for mTORC1 signaling both in normal and cancer cells and their potential as drug targets for cancer therapy.

## 2. Plasma Membrane AA Transporters Regulate mTORC1 Activity and Are Promising Therapeutic Targets

Transmembrane transporters (solute carriers, SLCs) are required to maintain intracellular homeostasis by allowing exchange of hydrophilic nutrients and metabolic waste across the plasma membrane of all cells. Among the SLCs, AA transporters promote the uptake (symporter) or the exchange (antiporter) of AAs using ion gradients (Na^+^, K^+^, H^+^ or Cl^−^) and/or secondary substrates. Consistent with the mTORC1 pathway dependency on AA homeostasis, multiple AA transporters have been shown to play a key role in promoting mTORC1 activation and cell proliferation in cancer (Figure 2) and thus represent a new class of promising therapeutic targets [16,17,18]. Among them, the large neutral amino acid transporter 1 (LAT1, *SLC7A5)* has received the greatest interest. LAT1 is a Na^+^-independent obligatory exchanger that promotes uptake of branched chain and bulky AAs that include the key family of essential (Val, Ile, Leu, Phe, Tyr, Trp) and conditionally essential AAs (Gln, Asn, and Met) combined with the efflux of glutamine or other members of the family [19]. LAT1 works as a heterodimer with its chaperone, the CD98 glycoprotein (heavy chain, SLC3A2), which promotes its stabilization and trafficking to the plasma membrane [15]. LAT1 overexpression has been reported in almost all cancer types [20,21]. This wide upregulation in tumors is explained by the fact that cancer cells, independently of their tissue of origin, rely on essential AA uptake. Therefore, to meet this demand, cancer cells are able to promote LAT1 expression through dysregulation of numerous transcription factors including HIF-2α, β-catenin, Myc, ATF4 and TEAD [22,23,24,25,26]. Work from our lab demonstrated that LAT1 transport activity is a key limiting step in cancer cell proliferation in vitro and in vivo by promoting leucine uptake and mTORC1 activity (Figure 2) [20]. Indeed, genetic or pharmacological disruption of LAT1 inhibits the proliferation of cancer cells from various origins by inducing an AA stress response and abolishing mTORC1 activity. Other studies have also shown that LAT1 inhibition can lead to apoptosis in certain types of cancer, such as T-cell acute lymphoblastic leukemia [27], osteosarcoma [28], cholangiocarcinoma [29] and thymic carcinoma [30]. These studies showing that LAT1 inhibition is an efficient anticancer strategy across many cancer types further increases the attractiveness of LAT1 as a therapeutic target and encourages continued development of specific inhibitors such as JPH203 [20,31].

Nevertheless, LAT1 controls mTORC1 activation in various normal tissues including the brain, skeletal muscle and the immune system. This raises the concern that the targeting of LAT1 could have toxic effects in patients. In the brain, LAT1 plays an important role during nervous system development and full body LAT1 deletion in mice is embryonic lethal and leads to robust neural defects correlated with a reduction of mTORC1 activity and cell proliferation [25]. In vitro, LAT1 is required for mTORC1 activity, dendrite maturation, and survival of bulb granule cell neurons [32].

In humans, homozygous missense mutations in the LAT1 gene have been linked to autism spectrum disorders and neurodevelopmental disorders characterized by microcephaly and seizures [33]. Since CNS specific Raptor deletion in mice causes microcephaly [34], it is possible that the brain size defect observed in patients with LAT1 mutations are due to a decreased mTORC1 activity. LAT1 also regulate mTORC1 activity in the skeletal muscle. Activation of mTORC1 in the muscle following an intraperitoneal leucine injection is blunted in mice with a skeletal muscle specific LAT1 deletion compared to control mice [35]. Moreover, when nutritionally challenged during fasting or under a reduced protein diet, these mice display a disrupted intracellular leucine concentration and decreased mTORC1 activity. However, despite having a clear role in mTORC1 activity in the muscle under certain conditions, LAT1 appears to be dispensable for the maintenance of muscle mass, potentially due to a functional redundancy with LAT2 that can sustain enough leucine uptake and mTORC1 activity [35]. Finally, through the control of mTORC1, LAT1 also regulates the differentiation and activation of multiple immune cell types. T-cell receptor-mediated induction of LAT1 is critical for T cell’s metabolic reprogramming [36,37]. Pharmacological blockade or loss of LAT1 inhibits T cell proliferation, differentiation/maturation and cytokine production. These defects are, at least partially, attributable to diminished uptake of leucine and mTORC1 signaling [37]. In human monocytes/macrophages, knock down or pharmacological inhibition of LAT1 during their activation decreases leucine uptake, mTORC1 activity, glycolysis, c-Myc level and therefore IL-1β production [38]. Similarly, LAT1 is required for mTORC1 mediated metabolic and functional responses in B-cells [39] and natural killer cells [40]. However, despite its role in regulating mTORC1 and multiple physiological processes in normal tissues, JPH203 was well tolerated in patients with advanced solid tumors during a phase I clinical trial [31]. It is possible that this tolerance of JPH203 in clinical trials is a result of an incomplete inhibition of LAT1. In adults, a small amount of net transport may suffice to maintain basal conditions. Clearly further studies are required for greater insights into the pharmacokinetics of this LAT1 inhibitor and its role at different stages of an individual’s lifespan.

The second AA transporter that has been linked to mTORC1 activity is the alanine-serine-cysteine transporter 2 (ASCT2, *SLC1A5*). ASCT2 is a Na^+^-dependent transporter that exchanges small neutral AAs (Ala, Ser, Cys, Gln, and Asn). Comparably to LAT1, ASCT2 has been shown to be overexpressed in a wide variety of cancers and has been proposed to be the main tumor-associated glutamine transporter [41,42]. Glutamine, as a glutamate precursor, is a key AA for tumor metabolism and proliferation by enabling production of energy and intermediate metabolites as well as mTORC1 activity (Figure 2) [11]. Indeed, ASCT2 knock down has been reported to strongly decrease mTORC1 activity and tumor growth in multiple xenograft models suggesting that ASCT2 may be, as well, a good therapeutic target for cancer therapy [43,44,45]. ASCT2 is expressed in a large panel of tissues including the brain, lung, skeletal muscle, immune system, kidney and adipose tissue. The loss of ASCT2 in mice is viable and does not give a particular phenotype under normal conditions suggesting that targeting ASCT2 may not be toxic for patients [46]. Interestingly, in addition to the direct sensing of glutamine by the mTORC1 pathway, a second indirect mechanism has been proposed to explain how glutamine activates mTORC1 and cancer cell proliferation. In 2009, Nicklin and colleagues suggested a model where LAT1 is functionally coupled to ASCT2. In this study, the authors demonstrated that cellular uptake of glutamine acts as a rate-limiting step in the leucine-dependent activation of mTORC1 [47]. Indeed, ASCT2 was shown to drive an increase in the intracellular concentration of glutamine, which is secondarily used as an efflux substrate by LAT1 to promote the uptake of extracellular leucine and therefore the activation of mTORC1. This concept that is now widely accepted and cited in the literature has been recently challenged by two independent approaches demonstrating that genetic disruption of ASCT2 does not decrease LAT1 transport activity nor mTORC1 levels in cancer cell lines [48,49]. This divergence has been explained by the functional redundancy with other glutamine transporters, such as the sodium-coupled neutral AA transporter 1 and 2 (SNAT1/2, *SLC38A1/2* [48]) (Figure 2). This concept of functional collaboration between AA transporters in the regulation of AA homeostasis and mTORC1 likely exists but the functional redundancy, the tissue-specific expression and the large spectrum of substrates of these carriers suggest that a single, unified AA transporter model that is common to every cell type/tissue does not exist. Therefore, unlike targeting LAT1, ASCT2 inhibition may not be an efficient therapeutic strategy across all cancer types due to functional redundancy with other glutamine transporters. However, ASCT2 is also able to promote the uptake of serine, another AA that is relevant in mTORC1 signaling and cancer. Indeed, serine can provide one-carbon precursors that are used in a variety of biosynthetic pathways including nucleotide synthesis, metabolites that have recently been demonstrated to be sensed by mTORC1 and which are essential for cell survival (Figure 2) [50]. Despite the fact that serine can be synthetized de novo, in certain contexts cancer cells become dependent on exogenous serine [51,52]. This dependency offers another potential vulnerability where ASCT2 inhibition may lead to mTORC1 inhibition and cancer cell death by inhibiting serine import. Besides, an important feature of the ASCT2 transport is that both transported AAs—glutamine and serine—are fundamental for the “redox code” of the cancer cell: serine-driven one-carbon metabolism aka folate cycle provides the cell with necessary reducing equivalents (NADPH) (Figure 2), while ASCT2-driven glutamine uptake fuels xCT-dependent import of the redox-active AA cysteine (explained extensively later (Figure 2) [53]). This is yet another situation where ASCT2 activity might influence mTORC1. Namely, the dependence of the mTORC1 on the redox state of the cell has been proposed many times, although the data collected up to now are still controversial. Earlier studies pointed out that UV radiation can induce phosphorylation and activation of mTORC1 in an [H_2_O_2_] concentration-dependent manner [54,55]. Also, it has been shown that a pro-oxidative environment is able to inactivate the TSC complex, allowing GDP-GTP conversion of Rheb and consequent activation of mTORC1 even under nutrient starvation conditions [56,57]. However, opposing effects of reactive oxygen species (ROS) on mTORC1 activity have also been reported. Namely, Alexander et al. showed that elevated levels of ROS induces activation of TSC tumor suppressor via ataxia-telangiectasia mutated (ATM) damage sensor residing in the peroxisomes [58]. In line with this are studies showing peroxisomal subcellular localization of the TSC1, TSC2 and Rheb in the liver cells, that respond to elevated level of ROS by inhibiting mTORC1 activity [59]. Although the precise effects of the redox state of the cell on mTORC1 remain to be illuminated, it is worth mentioning that AAs and AA transporters have to be incorporated into this equation, considering their central role in the maintenance of redox homeostasis [53].

The third AA transporter of great interest which shares the chaperon (CD98) with LAT1 is the Xc- system xCT (*SLC7A11*) [15,60]. It is a Na^+^-independent, Cl^−^-dependent AA exchanger that allows import of the oxidized form of cysteine, aka cystine (the dominant form of cysteine in the serum and culturing media), at the expense of glutamate efflux [60]. Cysteine is a non-essential AA as it can be synthesized in the cell from methionine via the transsulfuration pathway (TSP) [61]. This can explain a rather restricted expression of xCT transporters in normal organs and tissues such as the CNS, immune system and pancreas [62]. Also, whole body genetic ablation of xCT in mice did not show any profound effects on the physiology or morphology [63]. However, the expression of xCT has been detected in vitro in almost all cancer cell lines and numerous cancer types from patient samples [62]. This discrepancy in the expression profile of xCT has been explained by the semi-essential nature of certain AAs such as cysteine, serine or arginine. Namely, intracellular biosynthesis of these AAs under basal conditions satisfies the needs for normal cell function but becomes limiting once the metabolic needs of the cell increase, as is the case in cancer. The limitation of cysteine availability in tumors has been overcome, at least in part, by regulation of xCT expression by transcriptional factors frequently involved in the process of neoplastic transformation such as ATF4, NRF2, ETS1 and p53 [64,65,66,67,68,69]. The importance of the cyst(e)ine import by xCT in cancer has been highlighted by the breakthrough discovery of ferroptosis—the newly described type of non-apoptotic cell death. Namely, in 2012 the group from Columbia University demonstrated that xCT inhibition in RAS-mutated cancer cells leads to a specific type of cell death due to uncontrolled peroxidation of membrane lipids [70]. The logic behind this was that cysteine availability is the limiting step for biosynthesis of glutathione (GSH), which serves as a reducing power for the antioxidant enzyme glutathione peroxidase 4 (GPx4) involved in the removal of lipid peroxides [71]. However, accumulating data has since revealed that cysteine’s roles in ferroptosis goes well beyond GSH biosynthesis, including its role in synthesis of the Fe-S clusters, ubiquinol, maintenance of the ER and general redox homeostasis thanks to its free thiol group [72].

As previously mentioned, xCT seems dispensable for normal mTORC1 activity as the genetic invalidation of xCT in mice lacked an abnormal phenotype under homeostatic conditions [63]. On the contrary, the results obtained in our and other laboratories showed that inhibition or genetic invalidation of the xCT leads to an AA stress response and suppression of mTORC1 activity, proving the semi-essential nature of cysteine, as well as a context-dependent importance of xCT for mTORC1 activity [73]. Although no mTORC1 sensor for cysteine has been described, it is interesting to recall that methionine is indirectly sensed by mTORC1 through its metabolite SAM, an intermediate of TSP, which thus can serve as an indirect detector of cysteine as well (Figure 2) [8]. Also, as in the case of the ASCT2, xCT activity can influence mTORC1 activity via redox signaling, as cysteine represents a limiting AA for the synthesis of GSH, the main non-enzymatic antioxidant of the cell [74]. Besides, expression levels of xCT have been shown to influence sensitivity of cancer cells to glutamine starvation [75,76,77]. Namely, a large portion of cellular glutamate has been used by xCT for fueling cystine import. This, however, seems to limit cellular metabolic flexibility, making them more dependent on the glucose uptake for survival, while another essential nutrient in this context—glutamine, is deaminated and used by xCT. The collaboration of ASCT2 and xCT in both redox and metabolic contexts makes them promising candidates for anti-cancer drug development. In accordance with this hypothesis is the fact that genetic invalidation of the cystine transporter (xCT), as well as the serine transporter (ASCT2), in cancer cells leads to reduced/delayed tumor growth, while no effect has been observed in xCT or ASCT2 null mice under physiological conditions [49,73,78].

Arginine is a non-essential AA that can be synthetized from citrulline by the argininosuccinate synthetase (ASS). However, arginine auxotrophy is frequently observed in multiple cancer types due to a decreased expression of the ASS suggesting an essential role of arginine transporter for mTORC1 signaling in cancer [79]. The cationic amino acid transporter (CAT/SLC7 family) are the major arginine transporters in cells (Figure 2). These transporters mediate the Na^+^-independent uptake of cationic AAs (arginine, lysine, histidine). CAT-1 (*SLC7A1*) has been reported to be overexpressed in colorectal cancer [80], breast cancer cell lines [81], and chronic lymphocytic leukemia (CLL) [82]. CAT-1 knockdown in HG3 CLL, MCF-7 and T47D cells significantly reduced arginine uptake, abolished cell proliferation and impaired cell viability [81,82]. Surprisingly, despite the role of arginine as an mTORC1 activator, none of these studies have investigated the mTORC1 signaling in response to CAT-1 disruption and thus more work is required to evaluate if CAT-1 like LAT1 is a key transporter for mTORC1 signaling. Homozygous CAT-1 knockout mice die on the first day after birth because of anemia, suggesting that a therapeutic strategy targeting CAT-1 would be detrimental for patients [83]. Nevertheless, arginine deprivation is a therapeutic strategy that is currently investigated for arginine auxotroph cancers. Indeed, instead of targeting arginine uptake, efforts have been made in developing recombinant arginine degrading enzymes, including the recombinant human arginase (rhARG) [84] and the pegylated arginine deiminase (ADI-PEG20) [79,85]. Interestingly, treatment with rhARG reduces mTORC1 activity and induces cell death in NSCLC cells confirming the important role of arginine and potentially arginine transporters in sustaining mTORC1 activity in cancer (Figure 2) [86].

ATB^0,+^/*SLC6A14* has also been shown to regulate mTORC1 activity and tumor growth [87,88,89]. ATB^0,+^ is a unique carrier in that it transports nearly all (18 of the 20, except glutamate and aspartate) proteinogenic AAs and is obligatorily coupled to Na^+^ and Cl^-^ (Na^+^/Cl^−^/AA in a stoichiometry of 2:1:1) [18]. ATB^0,+^ is overexpressed in numerous cancer types including estrogen receptor-positive breast cancer, colon cancer, pancreatic cancer and cervical cancer [18]. While deletion of ATB^0,+^ in mice does not appear to have an effect on normal tissues, knockout of this transporter in two independent mouse models of spontaneous breast cancer abolishes tumor growth by disrupting AA homeostasis and decreasing mTORC1 activity [88]. In addition, knockdown or pharmacological inhibition decreased mTORC1 activity and growth of pancreatic cancer cells in mouse xenografts [87]. ATB^0,+^ is therefore an interesting candidate for the development of an anticancer therapy. However, even if it is clear that ATB^0,+^ plays a role in mTORC1 activity during carcinogenesis, the mechanism remains somewhat elusive. It is possible that ATB^0,+^ promotes the uptake of AAs essential for mTORC1 signaling. However, the fact that ATB^0,+^ appears to be expressed in the same cancer types as LAT1, ASCT2 and xCT, it is reasonable to predict that ATB^0,+^ may be part of a global functional coupling with these others transporters. Indeed, unlike these obligatory exchangers, ATB^0,+^ is a symporter and is able to accumulate its AA substrates in cancer cells unidirectionally (Figure 2) [89]. Thus, ATB^0,+^ may be providing exchangeable intracellular AAs for the other transporters allowing them to promote an AA uptake sufficient to sustain mTORC1 activity and the enhanced anabolism of cancer cells. Future studies are essential to determine how ATB^0,+^ controls mTORC1 activity in order to fully understand its potential in cancer therapy.

Plasma membrane transporters represent a promising class of therapeutic targets to potently inhibit AA uptake and therefore mTORC1 activity. However, as mentioned above, a subset of cancers, such as follicular lymphoma, appear to be mutated for the AA sensing branch of mTORC1 signaling and are therefore more resistant to AA limitation. This, however, does not mean that targeting AA transporters in these tumors would be inefficient. Even more, the inability of these tumors to downregulate mTORC1 following AA uptake inhibition may lead to a “metabolic catastrophe” due to sustained anabolic processes while nutrients are scarce. It would be interesting to investigate the potential of transporter inhibitors such as JPH203 in these tumor types.

## 3. Intracellular Transporters Regulate mTORC1 Activity

In 2005, Goberdhan and colleagues performed a genetic screen in fly (*D. melanogaster*) with the goal of finding new plasma membrane transporters linked to cell growth and mTORC1 activity. They identified CG3424 (*Path)* and CG1139, two carriers related to mammalian proton-assisted SLC36 amino acid transporters (PATs), as potent activators of mTORC1 and cell growth in vivo [90]. Indeed, mutations of *Path* resulted in flies with marked reduction in wing and eye size, whereas *Path* overexpression induced overgrowth of the differentiating eye. They further demonstrated that the effects of *Path* on cell and tissue growth are mediated by modulation of mTORC1 signaling. Later studies using knock down and overexpression of human PAT1 and PAT4 in HEK293 and MCF-7 breast cancer cells further demonstrated that PAT transporters are required for AA-dependent mTORC1 activation and cell proliferation [91,92]. These two human PATs can also promote growth in transgenic flies in vivo [91]. Interestingly, while the fly PAT CG1139 exhibits similar transport activities as mammalian PATs, CG3424 has been described as a low efficiency transporter. However, despite this weak ability to transport AAs, CG3424 is still able to activate mTORC1 when expressed in oocytes [90]. Therefore, this characteristic associated with the fact that PATs transporters are predominantly expressed in intracellular membranes such as the lysosome (PAT1) and the Golgi (PAT4) raised the idea that PATs may act as “transceptors” that regulate mTORC1 from inside the cell by sensing AA levels, possibly independent of their transport activity [93]. Interestingly, the glutamine transporter SNAT2 has also been described as a potential transceptor acting at the cell surface [94]. In their study, Pinilla and colleagues demonstrated that the non-metabolizable SNAT2 substrate MeAIB (α-methylaminoisobutyrate) could activate mTORC1 during AA starvation in L6 myotubes. In line with this observation, it has been reported that MeAIB treatment increases cell proliferation in MCF7 breast cancer and L6 myotubes.

Studies identifying the human PAT1 as an mTORC1 regulator appeared shortly after the identification of the lysosomal surface as the key place for mTORC1 regulation by AAs via the Rag GTPases (see part 1). In 2011, work from Sabatini’s group further suggested a v-ATPase-dependent lysosome-centric inside-out model of AA sensing by mTORC1 in which AAs must accumulate into the lysosome to activate mTORC1 [95]. In this study, the authors demonstrated that radiolabeled extracellular AAs enter the lysosome within 10 min following AA starvation, suggesting a role for lysosomal AA transporters in controlling mTORC1 in response to AAs. PAT1 was the first AA transporter located on the lysosomal surface, which was shown to interact physically with the Rag GTPases and to be required for the AA-dependent activation of mTORC1 (Figure 3) [96]. PAT1 has been proposed to be a part of the Ragulator-Rag GTPase sensing machinery that regulates mTORC1. More recently, a study suggested that PAT1 expression leads to reactivation of mTORC1 and drives an acquired resistance to CDK4/6 inhibitors [97]. However, the role of PAT1 in the regulation of mTORC1 remains somewhat controversial. Indeed, overexpression of PAT1 has also been shown to decrease mTORC1 activity in vitro [95]. Moreover, the substrate specificity of PAT1 does not match with the range of AAs known to activate mTORC1 at the lysosome. Therefore, more work is required to understand how PAT1 affects mTORC1 signaling and why some discrepancy exists in the literature. PAT4 is predominantly localized on the Golgi in several cell types and has been suggested to interact with Rab1A, Raptor, and mTOR. In HCT116 colorectal cancer cells, PAT4 promote a form of mTORC1 activity that is resistant to rapamycin treatment, as well as glutamine and serine withdrawal, while being insensitive to leucine levels [92]. These data support a second model in which an AA sensing machinery exists on the Golgi and regulates mTORC1 signaling in response to different subsets of AAs or cellular cues than the one at the lysosomal surface (Figure 3). It is, however, unclear if a functional relationship between PAT4 and the Arf1-dependent regulation of mTORC1 mentioned in the part 1 of this review exists, and therefore, more studies are required to clarify how different subcellular mechanisms of AA sensing may collaborate in the control of mTORC1 in response to a variety of AAs.

While no cystine/cysteine sensor has been described for mTORC1, the lysosomal cystine transporter, cystinosin, has been shown to interact with the Rag GTPases and potentially regulate mTORC1 [98]. Mutation in the gene *CTNS* that encodes for cystinosin is the main cause of the disease cystinosis, a rare autosomal recessive storage disorder characterized by defective lysosomal efflux of cystine. Using coimmunoprecipitation and mass spectrometry, Andrzejewska et al. demonstrated that cystinosin interacts with multiple members of the lysosomal AA sensing machinery, including the Ragulator complex, the Rag GTPases and the v-ATPase (Figure 3). They further demonstrated that proximal tubular cell lines derived from *Ctns^−/−^* mice display a reduced mTORC1 activity. Interestingly, reduction of the lysosomal cystine level using cysteamine did not rescue mTORC1 activity in these cells, suggesting that cystinosin itself regulates mTORC1 and may therefore be a cystine sensor. However, more work must be done to confirm and understand the mechanism by which cystinosin may control mTORC1 activity.

More recently, a third AA transporter, SLC38A9, has been shown to be part of the lysosomal branch of the AA sensing machinery for mTORC1 (Figure 3). SLC38A9 resides on the lysosomal membrane and acts as a direct sensor of arginine [99,100,101]. Two models have been proposed to explain how SLC38A9 stimulates mTORC1 in response to Arginine. First, a study from Sabatini’s group suggested a model where luminal arginine binds to SLC38A9 and directly stimulates its GEF activity toward the RagA/B, thereby promoting its GTP-loading state and mTORC1 recruitment to the lysosomal surface [99]. However, this model has been recently challenged by a study from the Hurley laboratory. Based on previous observation that RagA spontaneously exchanges GDP to GTP, Fromm and colleagues raised the question as to whether a GEF activity from SLC38A9 is required in this pathway. The lysosomal folliculin complex (LFC) consists of the inactive Rag GTPases dimer, the Ragulator, and the FLCN:FNIP2 (FLCN-interacting protein 2) GTPase activating protein complex and maintains the Rag GTPases in their inactive states during amino acid starvation [102]. Fromm and colleagues proposed a model where, in the presence of amino acids, the cytoplasmic tail of SLC38A9 destabilizes the LFC. This destabilization promotes the FLCN GAP activity toward RagC and allows the spontaneous nucleotide exchange by RagA, promoting, therefore, the active RagA-GTP/RagC-GDP states and therefore mTORC1 recruitment to the lysosome. It remains unclear if these two models co-exist or if one mechanism is dominant over the other one. Therefore, more work is required to fully understand how SLC38A9 regulates the nucleotide loaded states of the Rag GTPases, as well as to determine if these regulatory mechanisms can be extended to other transporters/transceptors interacting with the Rag GTPases.

In addition to the regulation of GEF and GAP, arginine binding to SLC38A9 also stimulates its transport activity, promoting the efflux of neutral AAs such as leucine from the lysosome [100]. Leucine released in the cytosol is sensed by the sestrin proteins, promoting RagA/B-GTP loading and mTORC1 activation. Stimulation of the lysosomal efflux activity enables the products of autophagic protein degradation and scavenging processes (macropinocytosis) to reactivate the mTORC1 pathway even when extracellular AAs are scarce. Indeed, SLC38A9 is required for macropinocytosis of albumin to activate mTORC1 and support growth of murine pancreatic KRAS^G12D/+^; P53^−/−^ tumors [100]. Furthermore, another study has suggested that SLC38A7 is the main lysosomal glutamine/asparagine exporter required for extracellular protein-dependent growth of cancer cells [103]. As SLC38A7 and SLC38A9 belong to the same family, it is reasonable to speculate that SLC38A7 can also act as a glutamine sensor to mTORC1. While it is strongly possible that glutamine released from the lysosome can activate mTORC1 in the cytoplasm (see part 1), it is highly unlikely that SLC38A7 exerts a GEF activity toward RagA/B since this transporter lacks a large part of the N-terminal cytosolic domain responsible for the interaction between SLC38A9 and the Rag-GTPases [103]. Therefore, while both contribute to intracellular AA homeostasis, only SLC38A9 appears to be part of the AA sensing machinery of mTORC1. However, these studies suggest that targeting lysosomal transporter could be an efficient anti-tumor strategy for cancers relying on macropinocytosis and lysosomal degradation of extracellular proteins such as pancreatic ductal adenocarcinoma (PDAC).

Lysosomal shuttling of the plasma membrane transporter LAT1 and/or its chaperone CD98 has also been suggested to regulate mTORC1 in response to AA availability (Figure 3) [104,105,106]. However, many discrepancies exist regarding both the mechanism of shuttling and how this affects mTORC1 signaling. In 2014, Milkereit et al. identified the lysosomal protein LAPTM4b as a binding partner for the protein complex LAT1/CD98 [104]. They showed that LAPTM4b recruits LAT1/CD98 to the lysosomal surface, promoting a flux of leucine into the lysosomes, and suggested that this lysosomal transport activity of LAT1 is required for mTORC1 activation via V-ATPase following leucine stimulation. Later in 2018, a second study from Weng et al. reported that girders of actin filaments (girdin) (also known as Gα-interacting vesicle-associated protein) interact with CD98 in an ERK- and AA signaling-dependent manner [105]. Mechanistically, in response to growth factors, ERK phosphorylates girdin on S233 and S237, while AAs, through an unknown mechanism, promote the ubiquitination of CD98 leading to a girdin-CD98 interaction. This interaction decreases the cell surface level of CD98 by promoting its endocytosis and shuttling to the lysosome. Surprisingly, unlike the one mediated by LAPTM4B, the authors reported that girdin-dependent lysosomal re-localization of CD98 disrupts the intracellular level of Gln and Leu and leads to a decreased mTORC1 activity, suggesting a potential negative feedback mechanism in response to growth factors and AA stimulation. Lastly, in 2019, Beaumatin et al. showed that the lysosomal protein DRAM-1, through its interaction with SCAMP3, can direct LAT1 and ASCT2 to the lysosomal surface [106]. However, unlike with LAPTM4B, the authors reported that lysosomal LAT1 and ASCT2 facilitate AA export to the cytoplasm and stimulate mTORC1. The number of discrepancies between these three studies, as well as those discussed previously for PAT1, highlight the difficulties of studying intracellular/organelle transporters in the context of mTORC1 signaling. Experimental conditions and design can strongly affect interpretations of the data. For example, unique cell lines metabolize AA differently and become addicted to a specific substrate such as glutamine or serine leading to potential differences in sensitivity/regulatory mechanisms for mTORC1 signaling. Moreover, experimental approaches used for AA starvation, AA stimulation and genetic manipulation of the transporters (acute vs. chronic knockout, overexpression, etc.) vary between studies. Indeed, it is possible that a chronic loss of a transporter leads to compensatory mechanism that may not exist in a wild type cell upon acute deregulation of the same AA transporter, while its overexpression may imbalance potential functional coupling with other transporters and result in abnormal accumulation/depletion of AA in certain intracellular compartment. Finally, potential redundancy as well as the fact that some of these transporters are both expressed at the plasma membrane and intracellularly make it challenging to interpret data when a whole cell system is used. On the other hand, utilization of isolated organelle does not reproduce the intracellular dynamics of the AA levels as well as the interconnection of the transport between plasma and the membranes of the other organelles that can also be a regulatory hub for mTORC1 (see part 1). Therefore, data need to be interpreted carefully knowing the limitations of the experimental system used. The biology of intracellular transporters remains in its infancy and the development of better in vitro and in vivo approaches will be required to assess the physiological and disease relevance of these transports not only for mTORC1 signaling, but for numerous other mechanism/signaling pathways.

## 4. mTORC1 Balances Supply and Demand by Promoting Amino Acid Uptake

In 2002, Edinger and colleagues demonstrated that through an unknown mechanism, the PI3K-Akt-mTORC1 pathway regulates surface expression of nutrient transporters [107]. More recent studies have shown that mTORC1, through a coordinated transcriptional response, promotes nutrient uptake required for the synthesis of macromolecules and growth.

The transcription factor Myc is one of the early genes transcriptionally induced in response to growth factors. Myc induces several hundred active loci controlling glycolysis, bioenergetics, ribosome biogenesis and nutrient supply among others [108]. In this context, it is not surprising that Myc is able to promote AA uptake by inducing the expression of both LAT1 and ASCT2 [23,109]. In addition to increasing glutamine uptake via regulation of ASCT2, Myc promotes glutaminolysis and anaplerosis. This mTORC1-dependent induction of Myc (Figure 4) leads to the anaplerotic entry of glutamine-derived carbon into the TCA cycle via regulation of the two enzymes glutaminase (GLS) and glutamate dehydrogenase (GDH) [109,110]. This mechanism replenishes the TCA cycle intermediates that are consumed as essential precursors for the synthesis of non-essential AAs, nucleotides, and lipids.

More recently, mTORC1 has been shown to regulate the protein expression of a second transcription factor: ATF4. ATF4, a member of the CREB/ATF family of bZIP transcription factors, is best known as the central mediator of the integrated stress response (ISR) [111]. The ISR is an adaptive signaling pathway where four stress-activated protein kinases (GCN2, PERK, HRI and PKR) sense a diverse range of stresses and converge on phosphorylation of the serine 51 on the eukaryotic translation initiation factor eIF2α. Phosphorylation of eIF2α leads to a general reduction in protein synthesis to preserve AA, energy and to overcome stresses such as ER unfolding protein stress response (UPR) [112]. In parallel, phosphorylated eIF2α up-regulates the translation of specific mRNAs including key transcription factors such as ATF4 and ATF5. Induction of ATF4 downstream of the ISR drives an adaptative transcriptional program promoting the expression of genes involved in non-essential AA biosynthesis and AA transport to mitigate stresses such as UPR stress or AA starvation [113]. In 2016, Ben Sahra and colleagues demonstrated that mTORC1 promote the protein expression of ATF4 independently of the ISR and phosphorylation of eIF2α [114]. In this study, the authors demonstrated that mTORC1 promotes de novo purine synthesis via the ATF4-dependent induction of the mitochondrial one-carbon metabolism enzyme MTHFD2. Later, Park et al. demonstrated that in addition to MTHFD2, mTORC1 transcriptionally regulates, via ATF4, the expression of AA transporters including LAT1/CD98, CAT-1 and xCT (Figure 4) [115]. The authors concluded that by controlling the expression of ATF4, mTORC1 balances AA supply with the demand for protein synthesis and other AA consuming processes (Figure 4). In line with these findings, a recent preprint from the Manning laboratory shows that the genes regulated by mTORC1-ATF4 represent the specific subset of ISR-ATF4-sensitive genes responsible for AA uptake, synthesis and tRNA charging, but also redox balance within the cell [116]. Namely, up-regulation of the xCT by the mTORC1-ATF4 axis stimulates cystine uptake and consequent GSH biosynthesis (see Plasma membrane transport section above). This, together with the previously mentioned induction of the main enzyme involved in the mitochondrial one-carbon metabolism, suggests that mTORC1 is able to orchestrate the cellular “redox code”, comprised of GSH and NADPH (Figure 3) [53], via ATF4 transcription activity in order to sustain growth.

Interestingly, another study also showed mTORC1-dependent induction of xCT in TSC2-KO or PTEN-KO cells, TSC1-KO mouse tissues and human kidney tumors, but the effect has been ascribed to mTORC1-driven up-regulation of the Oct1 transcription factor [117]. It is worth mentioning here that NRF2 might also be involved in xCT-induction in response to the mTORC1 activation. NRF2 is a master regulator of the enzymes required for glutathione synthesis, including xCT [64,66]. However, this pathway remains rather controversial, considering that different studies showed alternative effects of mTORC1 on NRF2, depending on the experimental settings and model system used [118,119]. Independently of the mechanism(s) by which mTORC1 induces expression of the xCT, this interconnection seems indisputable and could be explained by the fact that cysteine homeostasis is fundamental for normal functioning of the anabolic processes controlled by mTORC1 [120]. Namely, thanks to its highly reactive –SH group, cysteine represents the most conserved proteinogenic AA responsible for adequate synthesis and folding of the proteins [121]. Besides, as a rate-limiting AA for GSH synthesis, cysteine also plays a pivotal role in the maintenance of the redox balance necessary for the smooth running of all biosynthetic processes. As mentioned above, the great reliance of cancer cells on the import of cysteine for overall redox homeostasis reveals xCT as a vulnerability point and promising target for anti-cancer therapy, especially for highly metabolically active cancers. Indeed, our previous study on the PDAC cell line Capan-2 showed that treatment with transforming growth factor β (TGFβ) increases cellular sensitivity to xCT-inhibition, i.e., ferroptosis [73]. TGFβ has been a major inducer of epithelial-to-mesenchymal transition (EMT) during embryogenesis, cancer progression and fibrosis, but also promotes mTORC1 activity [122,123]. This was not the first time that EMT is related to higher sensitivity toward ferroptosis [124], but it was first time that increased mTORC1-activity has been suggested as an important factor in this phenomenon [73,125]. Accordingly, a new preprint from Conlon and colleagues revealed that ATP-competitive mTORC1, and mTORC1/PI3K inhibitors were associated with resistance to xCT-inhibition, suggesting that the efficacy of the ferroptosis inducers are dependent on mTORC1 activity [126]. Therefore, the efficiency of strategies targeting xCT could be significantly decreased if combined with inhibitors of mTORC1 in clinical settings. In addition, these studies demonstrate that mTORC1 balances AA supply and demand by promoting amino acid uptake and therefore sustaining other mTORC1-dependent anabolic processes.

## 5. Concluding Remarks and Perspectives

The increased dependency of rapidly growing tumors versus normal cells on a small set of AA transporters to sustain mTORC1 signaling and growth creates an attractive targetable vulnerability. This anti-cancer approach has demonstrated encouraging results, with many preclinical models uncovering specific AA transporters as promising targets [20,49,73,82,88]. Multiple AA transporter inhibitors have been developed and shown promising results in preclinical studies; however, high-affinity and selective inhibitors remain scarce and, to date, only one AA transporter inhibitor is currently in clinical trial [31]. The clinical relevance of AA transporters as therapeutic targets remains in its infancy. The recent studies aiming to solve the structure of AA transporters such as LAT1 [127], ASCT2 [128] and xCT [129] will help the development of new selective inhibitors and uncover the full potential of such therapeutic strategies. In addition to targeting plasma membrane carriers, future work should also investigate the therapeutic potential of targeting intracellular AA transporters. Indeed, previous publications have suggested that targeting lysosomal transporters could be an efficient anti-tumor strategy in PDAC [100,103].

Finally, the existence of adaptive and pro-survival pathways suggests that resistance to AA transport inhibitors will occur. Indeed, in response to amino acid starvation these pathways inhibit global protein synthesis [112,130] and promote autophagy [131,132], macropinocytosis [133] and the increased transcription/translation of specific stress response proteins such as AA transporters [113]. Future studies should anticipate and investigate the potential of combining amino acid transporter inhibitors with other therapeutic agent targeting these adaptative pathways.

## Figures and Tables

**Figure 1 ijms-22-00023-f001:**
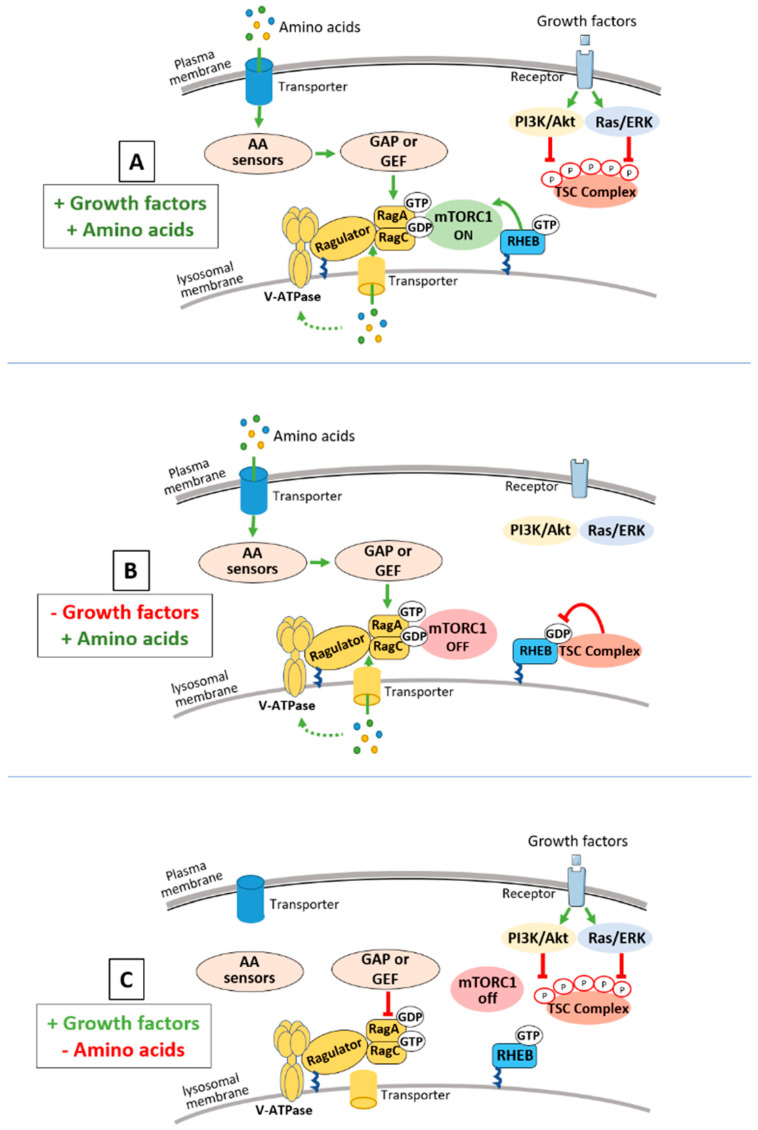
Mechanistic target of rapamycin complex 1 (mTORC1) integrates signals from growth factors and amino acids. This schematic represents different nucleotide loaded states of the Rheb and Rags GTPases in the presence or absence of amino acid (AA) and Growth factors and their influence on mTORC1 activity. (**A**) The presence of growth factors and hormones leads to an Akt- or Erk-dependent phosphorylation of tuberous sclerosis complex (TSC), resulting in the immediate release of the TSC complex from the lysosome promoting the accumulation of Rheb-GTP and therefore mTORC1 activation. (**B**) During a cellular stress or growth factors withdrawal, the TSC complex is recruited to the lysosome where it decreases Rheb-GTP levels and turns off mTORC1 signaling. AAs are sensed via a complex upstream network composed of AA sensors, AA transporters and GAP and GTP exchange factors (GEFs) factors that regulate the nucleotide loaded state of RagA/B and RagC/D. When AAs are present, the Rag heterodimer is converted to its active conformation, RagA/B-GTP and RagC/D-GDP, which leads to the lysosomal recruitment of mTORC1. (Panel **C**) When AAs are absent, the Rag heterodimer is converted to its inactive conformation RagA/B-GDP and RagC/D-GTP which promotes the release of mTORC1 into the cytoplasm (**C**). Signals from both AAs and growth factors/hormones are required to promote a full activation of mTORC1.

**Figure 2 ijms-22-00023-f002:**
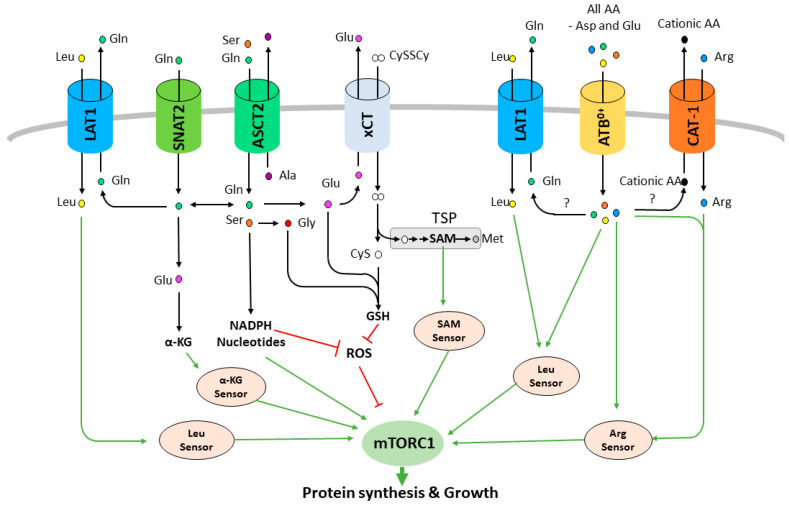
Amino acid transporters regulate mTORC1 activity. Amino acid transporters promote the uptake of leucine, arginine, glutamine, serine, cystine (CySSCy) and other amino acids. These amino acids are sensed directly to activate mTORC1 or converted to downstream metabolites that will activate mTORC1 or inhibit the production of reactive oxygen species (ROS). α-KG: α-ketoglutarate, SAM: S-adenosylmethionine, TSP: Transsulfuration pathway.

**Figure 3 ijms-22-00023-f003:**
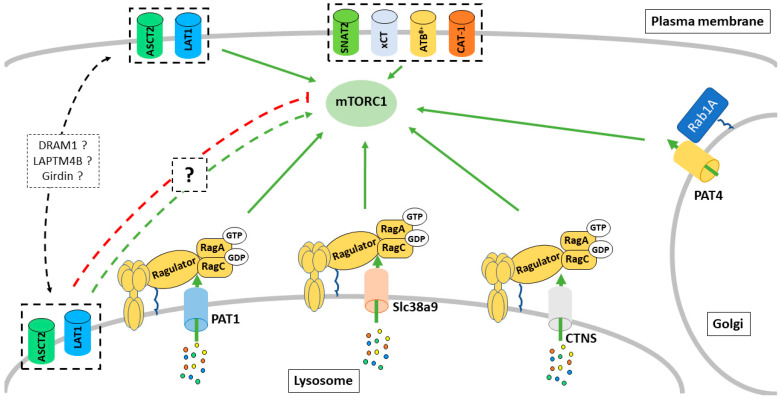
Intracellular amino acid transporters regulate mTORC1 activity. Multiple intracellular transporters have been shown to regulate mTORC1 via possibly Rag GTPase-dependent and independent mechanisms.

**Figure 4 ijms-22-00023-f004:**
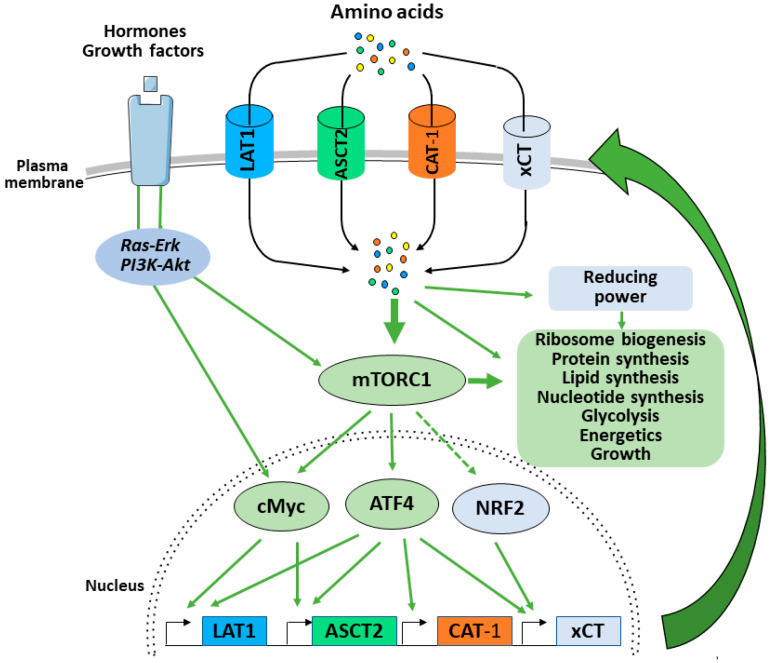
mTORC1 balances supply and demand by promoting amino acid uptake. When activated by growth factor, mTORC1 promotes the translation of cMyc, ATF4 and potentially stabilizes NRF2. These transcription factors will then promote the expression of AA transporters that will increase AA uptake, sustain mTORC1 activity and promote the mTORC1 mediated anabolic program and cell growth.

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
