# Peer review of "Amino Acid Transporters Are a Vital Focal Point in the Control of mTORC1 Signaling and Cancer"

_ijms, 2020, doi:10.3390/ijms22010023_

Round 1

Reviewer 1 Report

This is an excellent review of amino acid transporters and their role in mTORC1 activation in cancer cells. It highlights controversies and points to areas of limited understanding. References are up-to-date. My comments are minor.

Page 4, 2. Paragraph: The named essential amino acids contain real essential amino acids (Val, Ile, Leu, Phe/Tyr, Trp and conditionally essential AA Gln, Asn, Met). It is worth discirminating these.

Fig. 2 With the exception of ATB0,+ all depicted transporters are expressed widely in cancer cells. It might be worthwhile to indicate the limited distribution of ATB0,+ in cancer cells.

Page 6 1. Paragraph: The only explanation for the tolerance of JPH203 in clinical trials must be the incomplete inhibition of LAT1. In adults a small amount of net transport must suffice.

Page 8 Last paragraph: follicular lymphoma insensitive to AA starvation. Any cell must be sensitive to starvation of essential AA. It is the sensitivity to limited AA supply that must be different. It is still unresolved at what AA levels mTORC1 switches off. 

Page 10 3. Paragraph: Given that we have now a better structural understanding how transceptors activate mTORC1 (Fromm SA, Lawrence RE, Hurley JH. Structural mechanism for amino acid-dependent Rag GTPase nucleotide state switching by SLC38A9. Nat Struct Mol Biol. 2020 Aug 31. doi: 10.1038/s41594-020-0490-9. Epub ahead of print. PMID: 32868926.) this might be worth extending the paragraph.

Page 13 "...efficiency of strategies targeting xCT could be significantly decreased..." Shouldn't this be increased?

Author Response

Responses to reviewer 1 in bold-Italic

This is an excellent review of amino acid transporters and their role in mTORC1 activation in cancer cells. It highlights controversies and points to areas of limited understanding. References are up-to-date. My comments are minor.

We are grateful for positive feedback from the reviewer, and we hope that the alterations made to the manuscript will address all the minor issues satisfactorily.

Page 4, 2. Paragraph: The named essential amino acids contain real essential amino acids (Val, Ile, Leu, Phe/Tyr, Trp and conditionally essential AA Gln, Asn, Met). It is worth discriminating these.

According to this suggestion, discrimination between essential and conditionally-essential AA has been emphasized in the text.

Fig. 2 With the exception of ATB0,+ all depicted transporters are expressed widely in cancer cells. It might be worthwhile to indicate the limited distribution of ATB0,+ in cancer cells.

The cancer distribution is indicated in the text for each transporter.
Figure 2 aims to show how by transporting different substrates, these plasma transporters promote mTORC1 activity. Since this figure is already complex, we do not think adding the cancer distribution of these transporter is essential to this figure.

Page 6 1. Paragraph: The only explanation for the tolerance of JPH203 in clinical trials must be the incomplete inhibition of LAT1. In adults a small amount of net transport must suffice.

We are grateful for this referee`s remark. Indeed a small activity of LAT1, observed in ASCT2-KO cell, seems to be enough for maintenance of the amino acid homeostasis even in highly demanding cancer cells. Thereby, the hypothesis of JPH203 tolerance in clinical trail has been added to the text.

Page 8 Last paragraph: follicular lymphoma insensitive to AA starvation. Any cell must be sensitive to starvation of essential AA. It is the sensitivity to limited AA supply that must be different. It is still unresolved at what AA levels mTORC1 switches off. 

We agree with this remark and this has been changed in the text by “and are therefore more resistant to AA limitation”

Page 10 3. Paragraph: Given that we have now a better structural understanding how transceptors activate mTORC1 (Fromm SA, Lawrence RE, Hurley JH. Structural mechanism for amino acid-dependent Rag GTPase nucleotide state switching by SLC38A9. Nat Struct Mol Biol. 2020 Aug 31. doi: 10.1038/s41594-020-0490-9. Epub ahead of print. PMID: 32868926.) this might be worth extending the paragraph.

Thanks, this reference has been now added and discussed in the text

Page 13 "...efficiency of strategies targeting xCT could be significantly decreased..." Shouldn't this be increased?

This part refers to recent paper of Colon et al, who showed that mTORC1 inhibitors increase resistance to xCT-dependent ferroptosis. Thus, using them together indeed could lead to lower efficacy of the latter.

Reviewer 2 Report

This is a welll written and comprehensive review on the roles of amino acids transporters in regulating mTORC1 activity and the implications for cancer.  It provides a chronological look of this field covering activation of mTORC1 by both transport activity at the plasma membrane and also from intracellular compartments, where amino acid sensing versus transport has been highlighted. I have some points that I would liek to see addressed prior to pubication.

In Figure 1, I think it would be better to show amino acid transporters in the lysosomal membrane versus AA sensors in the cytoplasm. 

To make the review more balanced, in addition to showing specific amino acid transporters on the palsma membrane (Figure 2),  a figure including the specific intracellular transporters identified on the surface of late endosomal and lysosomal compartments, as well as those on the Golgi apparatus, is needed. 

The authors rightly point out the challenges of analysing intracellular amino acid transporters.  My view is that this review would, however, be strengthend by mention of the therapeutic potential and supporting studies of not only plasma mebrane transporters, but also intracellular transporters in the concluding section.

Formating points:

  1. It wold be more effective to have the panels in Figure 1 on one page, rather than split across two.
  2. Page 3, line 3:  'This' owuld sit better  with the rest of the sentence on the nest page..
  3. Page 6, line 22:  Suggest 'loss' rather than 'invalidation'

Author Response

Response to reviewer 2  in Bold Italic

This is a well written and comprehensive review on the roles of amino acids transporters in regulating mTORC1 activity and the implications for cancer.  It provides a chronological look of this field covering activation of mTORC1 by both transport activity at the plasma membrane and also from intracellular compartments, where amino acid sensing versus transport has been highlighted. I have some points that I would like to see addressed prior to publication.

We appreciate the referee’s positive outlook of our review. We have done our best to address all of the issues that have been pointed out by this referee as detailed below.

In Figure 1, I think it would be better to show amino acid transporters in the lysosomal membrane versus AA sensors in the cytoplasm. 

According to this suggestion, the figure has been updated with an AA transporter for the lysosomal part.

To make the review more balanced, in addition to showing specific amino acid transporters on the plasma membrane (Figure 2),  a figure including the specific intracellular transporters identified on the surface of late endosomal and lysosomal compartments, as well as those on the Golgi apparatus, is needed. 

We are grateful for this referee`s suggestion, as the importance of the transporters localized in the membrane of the intracellular organelles indeed deserves special attention in the future research. Thus, a new figure 3 has been added in the revised version of the manuscript.

The authors rightly point out the challenges of analysing intracellular amino acid transporters.  My view is that this review would, however, be strengthened by mention of the therapeutic potential and supporting studies of not only plasma membrane transporters, but also intracellular transporters in the concluding section.

In the Concluding remarks and perspective part, the therapeutic potential of targeting intracellular transporters has been highlighted.

Formating points:

  1. It would be more effective to have the panels in Figure 1 on one page, rather than split across two.

This has been changed

  1. Page 3, line 3:  'This' would sit better  with the rest of the sentence on the nest page..

This has been changed.

  1. Page 6, line 22:  Suggest 'loss' rather than 'invalidation'

This has been changed in the text

Round 2

Reviewer 2 Report

The authors have addressed my concerns and I am happy to now recommend publication.

Author Response

All changes indicated in the revised manuscript in RED COLOUR have now been checked and corrected in the final manuscript.

Ready for submission